# Cortical Thickness Changes in Migraine Patients Treated with Anti-Calcitonin Gene-Related Peptide Monoclonal Antibodies: A Prospective Age- and Sex-Matched Controlled Study

**DOI:** 10.3390/biomedicines13051150

**Published:** 2025-05-09

**Authors:** Soohyun Cho

**Affiliations:** Department of Neurology, Uijeongbu Eulji Medical Center, Eulji University School of Medicine, 712, Dongil-ro, Uijeongbu 11749, Republic of Korea; anttop@naver.com; Tel.: +82-31-951-1000; Fax: +82-2-974-7785

**Keywords:** migraine, treatment, cortical thickness, anti-calcitonin gene-related peptide monoclonal antibody, prospective study

## Abstract

**Background:** Migraine is associated with structural brain abnormalities, including cortical thickness changes. Anti-calcitonin gene-related peptide monoclonal antibodies (anti-CGRP mAbs) are a novel therapy for migraine prevention, but their effects on cortical structures are poorly understood. **Methods:** In this prospective age- and sex-matched controlled study, 30 migraine patients receiving either anti-CGRP mAbs (fremanezumab) (*n* = 15) or oral preventive medications (*n* = 15) underwent 3T MRI scans before and after treatment. Treatment response was defined as a ≥50% reduction in monthly headache days after 3 months. Cortical thickness was analyzed across 46 cortical regions, comparing patients treated with anti-CGRP mAbs to those receiving oral preventive treatment, as well as responders to non-responders within the anti-CGRP group. **Results:** Cortical thickness changes did not differ significantly between the anti-CGRP and oral treatment groups. However, among patients receiving anti-CGRP mAbs, responders showed significant decreases in cortical thickness compared to non-responders, particularly in the right caudal anterior cingulate (*p* = 0.026) and left rostral middle frontal cortex (*p* = 0.007). These cortical changes correlated with treatment response to anti-CGRP mAbs (β = −0.429, 95% CI [−0.777, −0.081], *p* = 0.016 in the right caudal anterior cingulate; β = −0.224, 95% CI [−0.390, −0.057], *p* = 0.008 in the left rostral middle frontal cortex). **Conclusions:** This exploratory study, based on a small sample size, suggests that cortical thickness changes may be associated with treatment response to anti-CGRP mAbs rather than with CGRP mAb treatment itself. Further studies with larger cohorts are needed to confirm these findings.

## 1. Introduction

Migraine is a common neurological disorder affecting approximately 14% of the global population, characterized by recurrent moderate to severe headaches often accompanied by autonomic and neurological symptoms [1]. Beyond its manifestations, migraine is increasingly recognized as a progressive neurological disorder with structural and functional brain abnormalities [2,3].

08Recent advances in migraine prevention include the development of monoclonal antibodies targeting the calcitonin gene-related peptide (CGRP) pathway [4]. These anti-CGRP monoclonal antibodies (mAbs) have demonstrated efficacy in reducing migraine frequency with favorable safety profiles [5]. While initially believed to act primarily through peripheral mechanisms by inhibiting CGRP-mediated vasodilation and neurogenic inflammation, emerging evidence suggests potential central effects of these medications [6].

Neuroimaging studies have demonstrated cortical thickness changes in migraine patients compared to healthy controls [7,8,9,10,11]. These changes primarily involve regions implicated in pain processing, including the somatosensory cortex, cingulate cortex, prefrontal regions, etc. A previous study suggested that these cortical abnormalities may correlate with clinical features such as disease duration, headache frequency, and psychological factors [12]. In addition, recent studies have shown that migraine treatments can influence cortical thickness [13,14,15,16]. For instance, prophylactic therapies and anti-CGRP monoclonal antibodies have been associated with changes in cortical regions, such as the posterior cingulate and anterior cingulate cortices, which correlate with clinical improvements [14]. This suggests that treatments may modulate brain structure in ways that reflect their therapeutic effects.

However, the effect of anti-CGRP mAbs on cortical thickness remains poorly understood, necessitating further investigation into their central mechanisms and potential as neuroimaging biomarkers of treatment response. This study aims to investigate cortical thickness changes in migraine patients treated with anti-CGRP mAbs compared to those receiving oral preventives. Additionally, it seeks to elucidate the associations between treatment response and cortical changes among patients treated with anti-CGRP mAbs, providing insights into their therapeutic effects and potential for personalized treatment approaches.

## 2. Methods

### 2.1. Patient Selection

In this prospective cohort study, we recruited patients with migraine who visited the Uijeongbu Eulji Medical Center in Seoul, South Korea, between January 2024 and December 2024. The study protocol received approval from the Uijeongbu Eulji Medical Center Institutional Review Board (UEMC 2023-09-013 and 2024-05-009). This study was not registered in a public trial registry. Inclusion criteria were as follows: (1) adults aged 18 to 65 years, (2) diagnosed with migraine (with or without aura) according to the International Classification of Headache Disorders, third edition (ICHD-3) criteria, (3) patients who were already on oral preventive medications could continue them throughout the study, and (4) patients who demonstrated good compliance with treatment and maintained a headache diary. All patients were evaluated by an experienced neurologist specializing in headache disorders (S.C.) and underwent brain imaging to rule out secondary headache disorders. Patients were excluded if they had hypertension, diabetes mellitus, dyslipidemia, cardiac disease, stroke, or a history of smoking.

### 2.2. Anti-CGRP Group and Oral TREATMENTGROUP

Participants were assigned to one of two groups: (1) the anti-CGRP mAbs treatment group (anti-CGRP group) or (2) the oral preventive medication treatment group (oral treatment group). The anti-CGRP group received fremanezumab administered subcutaneously either monthly or quarterly for a total duration of 3 months. Patients in the oral treatment group continued their existing oral preventive medications throughout the study period without any adjustments to the dose. To ensure comparability, the groups were matched for age and sex.

### 2.3. Study Protocol

At the initial visit, participants completed a structured self-reported questionnaire from the headache registry, which collected data on disease duration, headache frequency, headache severity, prodromal symptoms, and aura. All participants were followed up for a total of 3 months. Treatment response was assessed using monthly headache days (MHDs). Patients were classified as treatment responders if they achieved a ≥50% reduction in MHDs over the 3-month treatment period compared to baseline.

At Visit 1 (baseline), eligible patients underwent a baseline brain MRI scan and baseline MHDs were assessed through a neurologist-conducted interview. On the same day, participants in the anti-CGRP group received their first injection of fremanezumab, while those in the oral treatment group continued their prescribed oral preventive medication. Monthly follow-ups were conducted to assess treatment adherence and collect headache-related data.

At Visit 2: after 3 months of treatment, participants underwent a follow-up brain MRI scan. The same neurologist conducted a follow-up interview to assess changes in MHDs.

### 2.4. MRI Acquisition

We performed brain MRI scans using a 3T scanner (Magnetom Trio Tim, Siemens, Erlangen, Germany) with a 12-channel head coil. The imaging protocol included T1-weighted 3D magnetization-prepared rapid gradient echo (MPRAGE) sequences with the following parameters: TR = 2530 ms, TE = 3.45 ms, inversion time = 1100 ms, FOV = 256 mm, matrix = 256 × 256, section thickness = 1.0 mm, flip angle = 7 degrees, and voxel size = 1.3 × 1.0 × 1.3 mm.

### 2.5. Cortical Thickness Analysis

We analyzed 46 brain cortical regions. Cortical thickness measurements were performed using ATROSCAN, an AI-based brain segmentation software that utilizes convolutional neural networks. The processing pipeline and representative patient-level cortical maps are presented in Appendix A. Baseline T1-weighted 3D MPRAGE MRI images were processed through ATROSCAN’s algorithm, which includes preprocessing steps to adjust for white matter intensity variability. The software employs a 3D U-Net architecture to accurately delineate gray/white matter boundaries. ATROSCAN measurements showed a high correlation with FreeSurfer.

The ATROSCAN-generated cortical thickness measurements were validated against FreeSurfer, a widely used neuroimaging analysis tool, with a reported Pearson correlation coefficient of 0.9623, indicating high agreement between the two methods [17]. The robustness of ATROSCAN’s segmentation was further enhanced by applying data augmentation and preprocessing techniques to ensure reliable and reproducible cortical thickness measurements across different MRI acquisition settings.

### 2.6. Statistical Analysis

Baseline characteristics were summarized using descriptive statistics, presented as medians with interquartile ranges (IQRs) for continuous variables and frequencies with percentages for categorical variables. Group comparisons were performed using the Mann–Whitney U test for continuous variables, depending on the data distribution, and the chi-square test or Fisher’s exact test for categorical variables. Cortical thickness differences across 46 brain regions were analyzed using the Mann–Whitney U test, followed by false discovery rate correction to account for multiple comparisons. In addition to *p*-values, non-parametric effect sizes were estimated using Cliff’s delta to quantify the magnitude of group differences in cortical thickness changes. Cliff’s delta was interpreted based on conventional thresholds: |δ| < 0.147 as negligible, <0.33 as small, <0.474 as medium, and ≥0.474 as large. Subsequently, generalized linear models (GLMs) with a Gaussian distribution were used to identify clinical variables associated with changes in cortical thickness. The Shapiro–Wilk test confirmed that changes in cortical thickness followed a normal distribution. The dependent variable was the change in cortical thickness between the baseline and follow-up MRI, while the candidate independent variables included anti-CGRP mAbs treatment, treatment response to anti-CGRP mAbs, age, body mass index, chronic migraine, and baseline MHDs. To avoid overfitting, the number of predictors was limited based on clinical relevance and pre-specified hypotheses. Stepwise selection procedures were not employed. Multicollinearity among predictors was assessed using variance inflation factors, with all values below 2.0. Residuals were visually inspected and showed no major violations of model assumptions. Model fit was evaluated using deviance per degree of freedom, Akaike information criterion (AIC), corrected AIC (AICC), Bayesian information criterion (BIC), and consistent AIC (CAIC).

Statistical significance was set at *p* < 0.05. All statistical analyses were conducted using SPSS version 29.0 (IBM Corp., Armonk, NY, USA).

## 3. Results

### 3.1. Patient Characteristics

The study flowchart is shown in Figure 1. A total of 54 patients with migraine were recruited. Among them, 20 patients received anti-CGRP mAbs (anti-CGRP group), and 34 patients received oral preventive medication (oral treatment group). In the anti-CGRP group, four patients withdrew consent, and one patient did not complete the follow-up evaluation. In the oral treatment group, five patients withdrew consent, seven patients did not complete the follow-up evaluation, and seven patients were excluded due to age- and sex-matching criteria. Ultimately, 30 patients (15 in the anti-CGRP group and 15 in the oral treatment group) were included and analyzed in this study. Among the anti-CGRP group (n = 15), nine patients (60%) were responders to anti-CGRP mAbs, and six patients (40%) were non-responders.

The anti-CGRP group and oral treatment group exhibited comparable demographic and clinical profiles at baseline (Table 1). The median age was 47.0 years (range, 24–64 years) in the anti-CGRP group and 44.0 years (range, 23–64 years) in the oral treatment group (*p* = 0.645). All participants were female. Chronic migraine was reported in 53.3% of the anti-CGRP group and 40.0% of the oral treatment group (*p* = 0.330). Baseline MHDs were not significant between groups, with a median of 21.0 days in the anti-CGRP group and 12.0 days in the oral treatment group (*p* = 0.312).

### 3.2. Comparison Between Anti-CGRP Group and Oral Treatment Group

While median MHDs decreased in both groups after three months, the anti-CGRP group had a significantly higher response rate (57.1% vs. 12.5%, *p* = 0.035) and a greater proportion of ≥50% responders (60.0% vs. 13.3%, *p* = 0.021). A full list of regional cortical thickness changes is provided in Appendix A. Across 46 cortical regions, including the right caudal anterior cingulate cortex and left rostral middle frontal cortex, no significant differences in cortical thickness were observed between the groups (Table 2 and Appendix A). In the right caudal anterior cingulate cortex, the anti-CGRP group showed a median cortical thickness reduction of −0.120 mm, while the oral treatment group exhibited an increase of 0.400 mm (*p* = 0.560, Cliff’s delta = 0.293). In the left rostral middle frontal cortex, both groups showed cortical thickness reduction, with a median change of −0.120 mm in the anti-CGRP group and −0.300 mm in the oral treatment group, which was not significant between groups (*p* = 0.675, Cliff’s delta = 0.229).

### 3.3. Comparison Between Responders and Non-Responders to Anti-CGRP mAbs

Among 46 cortical regions, significant cortical thickness changes were observed in two areas (Table 3 and Figure 2). In the right caudal anterior cingulate cortex, responders showed a significant reduction in cortical thickness (−0.170 mm), while non-responders exhibited an increase (0.130 mm, *p* = 0.026, Cliff’s delta = 0.704). After three months, cortical thickness remained lower in responders (2.990 mm) than in non-responders (3.450 mm, *p* = 0.012). A similar pattern was observed in the left rostral middle frontal cortex. Responders showed a significant reduction in cortical thickness (−0.170 mm), whereas non-responders had an increase (0.055 mm, *p* = 0.007, Cliff’s delta = 0.815). The post-treatment cortical thickness difference between groups did not reach statistical significance (*p* = 0.082).

### 3.4. Association Between Clinical Variables and Cortical Thickness Changes

Generalized linear models identified significant associations between treatment response to anti-CGRP mAbs and cortical thickness changes in specific brain regions (Table 4). In the right caudal anterior cingulate cortex, treatment response to anti-CGRP mAbs exhibited significantly greater reduction of cortical thickness compared to non-responders (β = −0.429, 95% CI: −0.777 to −0.081, *p* = 0.016). The GLM for the right caudal anterior cingulate cortex showed a deviance per degree of freedom of 2.959, with AIC = 31.646, AICC = 38.503, BIC = 42.856, and CAIC = 50.856. Similarly, in the left rostral middle frontal cortex, responders showed significant reduction of cortical thickness (β = −0.224, 95% CI: −0.390 to −0.057, *p* = 0.008). The GLM for the left rostral middle frontal cortex demonstrated a deviance per degree of freedom of 0.674. Model fit indices were as follows: AIC = –12.735, AICC = –5.878, BIC = –1.526, and CAIC = 6.474. Other clinical variables, including age, BMI, chronic migraine status, baseline MHDs, and anti-CGRP treatment itself, were not significantly associated with cortical thickness changes in either region.

## 4. Discussion

This prospective study investigated cortical thickness changes in migraine patients treated with anti-CGRP monoclonal antibodies compared to those receiving oral preventives, along with exploring associations between treatment response and structural brain changes. Our findings indicate that (1) anti-CGRP mAbs treatment itself did not lead to region-specific cortical thickness changes compared to oral preventives; (2) however, among anti-CGRP mAbs recipients, responders exhibited cortical thickness reduction in specific regions, particularly the right caudal anterior cingulate and left rostral middle frontal cortex; and (3) in GLM analysis, treatment response to anti-CGRP mAbs was significantly associated with cortical thickness reduction in these regions.

This study demonstrated that anti-CGRP mAbs treatment itself did not lead to significant cortical thickness changes compared to oral preventive medications. While the study was adequately powered for exploratory comparisons and initial hypothesis generation, it was not designed to detect subtle differences with high statistical precision. This limitation should be considered when interpreting non-significant findings between-group comparisons. Although the sample size was sufficient for exploratory analysis, we acknowledge that it may have limited the ability to detect subtle but clinically relevant cortical changes. The absence of statistically significant group-level differences may, therefore, reflect insufficient power rather than a true lack of effect.

An additional challenge in interpreting the between-group results is the heterogeneity of medications used in the oral treatment group. The oral treatment group included a variety of preventive agents, such as topiramate, beta-blockers, and antidepressants, each of which may exert distinct effects on brain structure. Although our results did not show significant cortical differences between the anti-CGRP and oral treatment groups, we cannot rule out the possibility that these pharmacological differences may have contributed to variability in structural outcomes. Future studies with more homogeneous control arms or medication-specific subgroup analyses will be necessary to better delineate treatment-related brain effects.

In contrast, cortical thickness reductions were observed only in responders to anti-CGRP mAbs treatment. GLM analysis indicated that treatment response was significantly associated with cortical thickness reduction in specific brain regions, whereas anti-CGRP mAbs administration itself did not show a statistically significant association. The cortical thinning observed in responders, especially in pain-processing regions such as the anterior cingulate and rostral middle frontal cortex, raises important questions regarding its biological significance. One plausible interpretation is that this reflects functional normalization of previously altered cortical areas. Several studies have reported that migraineurs exhibit cortical thickening in pain-related areas, likely due to ongoing nociceptive input and maladaptive neuroplastic changes [7,8]. Effective treatment may reduce this input and induce cortical reorganization toward a more normalized structure. Alternatively, these findings may represent adaptive plasticity, such as synaptic pruning or modulation of gray matter density associated with symptom relief, as observed in other chronic pain conditions [18]. It is important, however, to distinguish such dynamic reorganization from pathological atrophy, which typically accompanies neurodegeneration and is unlikely to occur over a short treatment duration in otherwise healthy individuals. Given the 3-month timeframe and lack of clinical deterioration in our cohort, atrophy is unlikely to explain the observed findings. Functional imaging studies further showed that effective migraine treatment can modulate activity in the anterior cingulate and prefrontal cortex, regions involved in both pain affect and top-down modulation [13,19,20]. Our structural findings may reflect similar treatment-induced adaptations at the morphometric level. Nevertheless, further longitudinal and multimodal studies are needed to clarify whether the observed changes represent normalization, plasticity, or treatment-induced reorganization.

The mechanism of action of anti-CGRP mAbs primarily involves a peripheral blockade of CGRP-mediated vasodilation and neurogenic inflammation [4]. Additionally, their large molecular size limits blood–brain barrier penetration [4]. Based on these findings, we propose a sequential mechanism in which the successful peripheral blockade of CGRP signaling reduces nociceptive input to central pain-processing networks, ultimately leading to cortical reorganization. This response-dependent structural change underscores the potential for anti-CGRP mAbs to modulate migraine-related brain abnormalities indirectly through their peripheral effects.

While the treatment modality itself did not produce differential cortical thickness changes, our analysis of responders versus non-responders within the anti-CGRP mAbs group revealed that responders to anti-CGRP mAbs treatment exhibited significant cortical thickness reductions in two regions: the right caudal anterior cingulate cortex and the left rostral middle frontal cortex. In contrast, non-responders showed cortical thickness increases in these regions. The observed pattern of cortical thickness changes with anti-CGRP mAbs treatment aligns with previous neuroimaging findings in migraine [14,16]. The anterior cingulate cortex is one of component of the pain-processing network and has been consistently implicated in migraine pathophysiology [21,22]. This region shows functional activation during migraine attacks and structural abnormalities in migraine patients compared to healthy controls [22]. Our finding of treatment-associated reduction in anterior cingulate cortical thickness is consistent with the results reported by Szabó et al. who observed decreased thickness in the anterior cingulate cortex following galcanezumab treatment [14]. In another study, functional MRI studies showed erenumab responders exhibited altered pain-induced activations in middle and posterior cingulate regions and periaqueductal gray matter [13]. The rostral middle frontal cortex, part of the prefrontal cortex, also plays a role in pain modulation and cognitive aspects of pain experience [23,24]. Abnormalities in this region have been associated with migraine chronification and disability [25]. The reduction in cortical thickness in this region following successful anti-CGRP mAbs treatment may reflect normalization of altered pain processing networks.

The variability in individual response to anti-CGRP mAbs may be attributed to several biological factors. Genetic differences such as polymorphisms in the CGRP gene may influence ligand–receptor binding affinity. Additionally, the development of neutralizing antibodies against monoclonal antibodies could attenuate therapeutic efficacy in some patients. Although these mechanisms remain speculative, they underscore the need for further research to identify predictors of treatment response and to guide personalized migraine therapy.

Previous studies on other migraine preventive treatments have reported cortical changes in different regions than those observed in our study. Hubbard et al. found that onabotulinumtoxinA treatment increased cortical thickness in several regions, including the right primary somatosensory cortex and anterior insula, in treatment responders [26]. Amaral et al. reported that routine oral preventive treatment reduced cortical thickness in the left posterior cingulate, which correlated with headache index improvement [16]. Additionally, sphenopalatine ganglion block treatment led to a significant decrease in cortical thickness in the left temporal pole and left lateral occipitotemporal gyrus [15]. These findings imply the specificity of cortical changes associated with different treatment approaches, suggesting that structural brain changes may be treatment-specific rather than indicative of a common pathway for migraine improvement.

The association between treatment response and cortical thickness changes has several important clinical implications. Our findings suggest that cortical thickness measurements may potentially serve as neuroimaging biomarkers of treatment response to anti-CGRP mAbs. This could help identify patients most likely to benefit from these treatments and allow for more personalized treatment approaches. The finding that cortical changes correlate with clinical improvement rather than treatment administration supports the use of clinical response as the primary outcome measure in migraine treatment studies. It also suggests that central effects of anti-CGRP mAbs are likely secondary to their peripheral mechanism of action, reinforcing the importance of peripheral CGRP signaling in migraine pathophysiology. Nevertheless, our study has several limitations. First, the relatively small sample size may have limited our ability to detect cortical changes. Second, the relatively broad age range of 18 to 65 years among participants poses a challenge, as cortical thickness naturally varies with age. This may have influenced the observed differences in cortical thickness, limiting the generalizability and interpretability of the findings. Third, although we utilized age- and sex-matched migraine patients naïve to CGRP mAb treatment, lack of normative cortical thickness data limits the interpretation of whether the observed changes reflect normalization or are treatment-induced. Fourth, the exclusively female sample precludes assessment of sex-specific treatment effects, despite migraine’s female predominance. Fifth, in our study, the oral treatment group’s continued use of varied oral preventives introduces potential confounding from polypharmacy effects. However, we could not control the oral preventive treatment type due to the ethical problem. Sixth, the 3-month follow-up period may not have been sufficient to capture the full extent of cortical reorganization following treatment. Seventh, while we applied non-parametric statistical tests and false discovery rate corrections, the small sample size and group stratification inevitably raise the possibility of type II errors. Effect size estimates using Cliff’s delta revealed small effects between the anti-CGRP treatment and oral treatment groups, suggesting potential biological relevance that may not have reached statistical significance due to limited statistical power. Future studies should include larger cohorts and longer follow-up periods to validate our findings and investigate whether these cortical changes persist or evolve over time. Lastly, the small number of patients receiving each type of oral preventive medication limited our ability to perform subgroup or sensitivity analyses based on drug class. As a result, we could not assess the potential structural brain effects of individual drugs separately. This heterogeneity in the oral treatment group may represent a source of confounding and should be considered when interpreting the findings.

## 5. Conclusions

This exploratory study suggests that cortical thickness changes may be more closely associated with treatment response to anti-CGRP mAbs than with the administration of anti-CGRP mAbs themselves. These findings suggest that anti-CGRP mAbs may modulate migraine-related brain abnormalities in responders, providing insights into their central mechanisms of action beyond peripheral effects. However, given the small sample size and the heterogeneity of oral preventive treatments used in the control group, these findings should be considered preliminary and require validation in larger, more homogeneous cohorts.

## Figures and Tables

**Figure 1 biomedicines-13-01150-f001:**
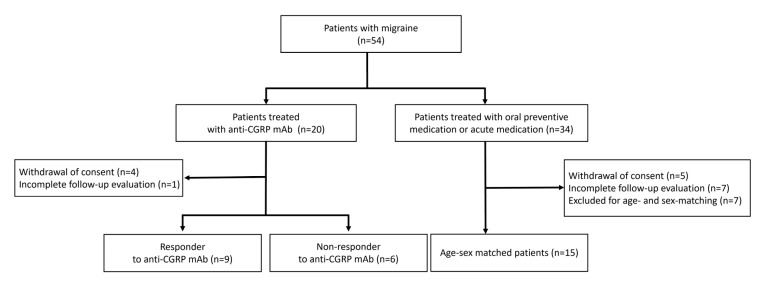
Flow diagram of the study.

**Figure 2 biomedicines-13-01150-f002:**
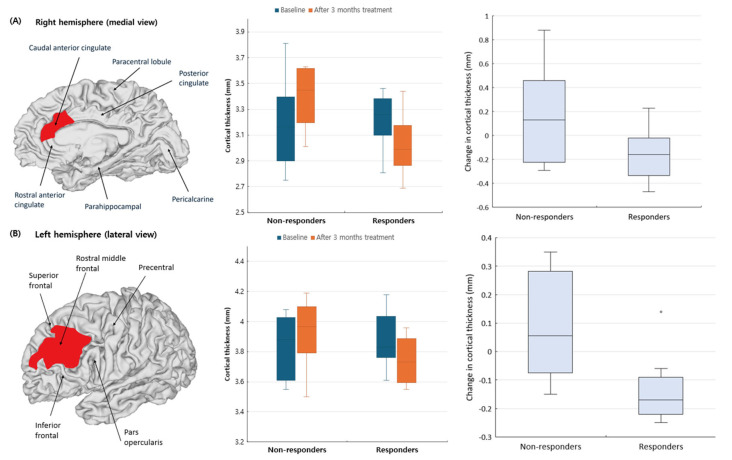
Cortical thickness changes in anti-CGRP responders versus non-responders. (**A**) The right caudal anterior cingulate cortex showing significant cortical thinning in responders compared to non-responders (**B**) The left rostral middle frontal cortex showing significant reduction in cortical thickness in responders. Brain surfaces are displayed with anatomical labels and group differences in cortical thickness are visualized.

**Table 1 biomedicines-13-01150-t001:** Comparison of baseline characteristics and clinical outcomes between anti-CGRP group and oral treatment group.

	Anti-CGRP Group (n = 15)	Oral Treatment Group * (n = 15)	*p*-Value
Age, years	47.0 (34.0–52.0)	44.0 (34.0–51.0)	0.645
Female sex	15 (100.0%)	15 (100.0%)	>0.999
BMI, kg/m^2^	23.4 (22.4–25.1)	21.5 (20.6–23.8)	0.021
Disease duration, months	120 (96.0–240.0)	120 (48.0–180.0)	0.451
Aura, n	2 (13.3%)	3 (20.0%)	0.165
Prodromal symptoms, n	14(93.3%)	11 (73.3%)	0.464
Chronic migraine, n	8 (53.3%)	6 (40.0%)	0.330
Baseline MHDs, days	21.0 (8.0–30.0)	12.0 (8.0–30.0)	0.312
Headache severity, NRS	8.0 (7.0–9.0)	7.0 (5.0–8.0)	0.007
Oral preventive medication use, n			
Amitriptyline	4 (26.7%)	1 (6.7%)	0.330
Topiramate	4 (26.7%)	11 (73.3%)	0.027
Beta blocker	1 (6.7%)	7 (46.7%)	0.035
Calcium channel blocker	0 (0.0%)	0 (0.0%)	
Divalproex sodium	0 (0.0%)	0 (0.0%)	

* Oral preventive group served as the comparator group (oral treatment group).

**Table 2 biomedicines-13-01150-t002:** Comparison of clinical outcomes between anti-CGRP group and oral treatment group.

	Anti-CGRP Group (n = 15)	Oral Treatment Group (n = 15)	*p*-Value
Baseline MHDs, days	21.0 (8.0–30.0)	12.0 (8.0–30.0)	0.312
MHDs after 3 months, days	4.0 (3.0–14.0)	9.0 (3.0–22.0)	0.453
Response rate to treatment	57.1% (0.0–84.6)	12.5% (0.0–40.0)	0.035
≥50% responder to treatment	9 (60.0%)	2 (13.3%)	0.021
MRI follow-up duration, days	91.0 (91.0–96.0)	91.0 (91.0–92.0)	0.430
Cortical thickness of right caudal anterior cingulate cortex, mm			
Baseline	3.240 (3.080–3.310)	3.150 (3.010–3.500)	0.830
After 3 months treatment	3.140 (2.920–3.440)	3.230 (3.090–3.370)	0.645
Change in cortical thickness	−0.120 (−0.290–0.180)	0.400 (−0.220–0.150)	0.560
Cortical thickness of left rostral middle frontal cortex, mm			
Baseline	3.830 (3.750–4.010)	3.810 (3.710–3.930)	0.372
After 3 months treatment	3.860 (3.610–3.960)	3.770 (3.640–3.860)	0.372
Change in cortical thickness	−0.120 (−0.210–0.120)	−0.300 (−0.150–0.050)	0.675

**Table 3 biomedicines-13-01150-t003:** Comparison of clinical outcomes between responders and non-responders to anti-CGRP mAbs.

	Responders (n = 9)	Non-Responders (n = 6)	*p*-Value
Baseline MHDs, days	26.0 (14.0–30.0)	17.5 (5.3–28.5)	0.227
MHDs after 3 months, days	4.0 (3.0–5.5)	14.5 (6.8–28.5)	0.017
Response rate to treatment	81.0% (58.6–88.3)	0.0% (0.0–34.2)	<0.001
MRI follow-up duration, days	91.0 (91.0–97.0)	91.0 (87.5–95.3)	0.405
Cortical thickness of right caudal anterior cingulate cortex, mm			
Baseline	3.260 (3.100–3.380)	3.160 (2.900–3.398)	0.438
After 3 months treatment	2.990 (2.865–3.175)	3.450 (3.198–3.615)	0.012
Change in cortical thickness	−0.170 (−0.335–−0.110)	0.130 (−0.005–0.460)	0.026
Cortical thickness of left rostral middle frontal cortex, mm			
Baseline	3.830 (3.760–4.035)	3.880 (3.610–4.028)	0.887
After 3 months treatment	3.730 (3.595–3.885)	3.965 (3.793–4.100)	0.082
Change in cortical thickness	−0.170 (−0.220–−0.100)	0.055 (−0.068–0.283)	0.007

**Table 4 biomedicines-13-01150-t004:** Generalized linear models of clinical variables related to change in cortical thickness.

	Right Caudal Anterior Cingulate Cortex			Left Rostral Middle Frontal Cortex		
	B (95% CI)	SE	*P*	B (95% CI)	SE	*P*
Intercept	0.061 (−0.986–1.108)	0.534	0.909	0.075 (−0.425–0.575)	0.255	0.769
Age	−0.002 (−0.013–0.009)	0.006	0.739	−0.002 (−0.007–0.004)	0.003	0.550
BMI	−0.003 (−0.041–0.034)	0.019	0.855	−0.003 (−0.021–0.015)	0.009	0.750
Chronic migraine	−0.134 (−0.457–0.189)	0.165	0.416	−0.002 (−0.156–0.152)	0.079	0.977
Baseline MHDs	0.007 (−0.009–0.024)	0.008	0.374	0.001 (−0.007–0.009)	0.004	0.808
Anti-CGRP treatment	0.259 (−0.045–0.563)	0.155	0.095	0.134 (−0.011–0.279)	0.074	0.070
Response to anti-CGRP mAbs	−0.429 (−0.777–−0.081)	0.178	0.016	−0.224 (−0.390–−0.057)	0.085	0.008

## Data Availability

The original contributions presented in this study are included in the article. Further inquiries can be directed to the corresponding author.

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
