# Peer review of "Cortical Thickness Changes in Migraine Patients Treated with Anti-Calcitonin Gene-Related Peptide Monoclonal Antibodies: A Prospective Age- and Sex-Matched Controlled Study"

_biomedicines, 2025, doi:10.3390/biomedicines13051150_

Round 1
Reviewer 1 Report
Comments and Suggestions for Authors
This manuscript presents a prospective, controlled, age- and sex-matched study examining cortical thickness changes in migraine patients treated with anti-CGRP monoclonal antibodies. The authors compare responders and nonresponders to treatment, as well as patients receiving standard oral prophylaxis. The topic is clinically relevant and timely, particularly given the current interest in discovering central nervous system correlates of peripheral migraine treatments.
Overall, the manuscript is well written and structured, and the use of image analysis techniques to explore cortical thickness as a marker of treatment response is commendable. However, several key issues related to sample size, group comparability, statistical interpretation, and methodological transparency need to be addressed to improve the robustness and clarity of the findings.
Major comments
The overall sample size (n = 30, with 15 per group) is a critical limitation, compounded by the stratification of the treatment group into responders and nonresponders. In such small samples, the risk of type II error is substantial, and the absence of significant results between the treatment and control groups may be due more to insufficient power than to a true lack of effect. I strongly recommend that the authors address this limitation earlier in the manuscript (it is currently only briefly mentioned) and consider including a post hoc power analysis or estimated effect sizes to contextualize the observed associations.
The control group consisted of patients taking a variety of oral preventive drugs, including medications with different mechanisms (e.g., topiramate, beta-blockers, and antidepressants). This heterogeneity may confound comparisons between groups, especially if some treatments have structural brain effects. Although the limitations section addresses this issue, the implications are not fully understood. I recommend expanding this discussion and, if possible, performing sensitivity analyses (e.g., excluding beta-blockers) or subgroup analyses based on drug class to assess the robustness of the findings.
The finding that patients who respond to anti-CGRP mAbs show cortical thinning in specific regions is intriguing, but interpretation remains speculative. It is unclear whether these changes reflect functional recovery, normalization of previously altered structures, or possible atrophy. The current discussion briefly references previous studies, but I believe this section would benefit from a more in-depth analysis of the possible mechanisms underlying the observed changes in cortical structure, particularly in pain-related regions such as the anterior cingulate and prefrontal cortex.
The use of generalized linear models (GLMs) is appropriate, but given the small sample size, there is a high risk of overfitting. The authors should clarify how they checked assumptions (e.g., multicollinearity, residual analysis) and whether model selection strategies (e.g., AIC, BIC) were employed. Parsimonious modelling may be more appropriate in this context. Additionally, p values close to the significance threshold (e.g., p = 0.070) should be interpreted with caution and not overemphasized in the narrative.
The use of ATROSCAN for cortical thickness analysis is a novel element of the study, but its application and validation in this specific context are not sufficiently explained. FreeSurfer remains the most widely used tool in this field, so it is essential to briefly justify the choice of ATROSCAN, the reference validation studies and comment on its comparability in terms of reliability and sensitivity, particularly for detecting subtle changes in cortical morphology.
Minor comments
Abstract: Please clarify in the abstract that only significant cortical differences were observed among responders within the anti-CGRP group, and that no significant differences were found in the overall comparison with the control group.
Terminology: Ensure consistent use of terms such as “treatment group” versus “anti-CGRP group” throughout the manuscript.
Figures: Figure 2 would benefit from clearer anatomical labels and a legend indicating the direction and magnitude of cortical changes.
Language: There are several minor grammatical issues (e.g., “an increase” should be “an increase”) that should be addressed with a final revision of the language.
Author Response
Reviewer 1
This manuscript presents a prospective, controlled, age- and sex-matched study examining cortical thickness changes in migraine patients treated with anti-CGRP monoclonal antibodies. The authors compare responders and nonresponders to treatment, as well as patients receiving standard oral prophylaxis. The topic is clinically relevant and timely, particularly given the current interest in discovering central nervous system correlates of peripheral migraine treatments.
Overall, the manuscript is well written and structured, and the use of image analysis techniques to explore cortical thickness as a marker of treatment response is commendable. However, several key issues related to sample size, group comparability, statistical interpretation, and methodological transparency need to be addressed to improve the robustness and clarity of the findings.
Author response: We sincerely thank the reviewer for the thoughtful and constructive comments on our manuscript. We have revised the manuscript where necessary to clarify the reviewer’s comments.
Major comments
- The overall sample size (n = 30, with 15 per group) is a critical limitation, compounded by the stratification of the treatment group into responders and nonresponders. In such small samples, the risk of type II error is substantial, and the absence of significant results between the treatment and control groups may be due more to insufficient power than to a true lack of effect. I strongly recommend that the authors address this limitation earlier in the manuscript (it is currently only briefly mentioned) and consider including a post hoc power analysis or estimated effect sizes to contextualize the observed associations.
Author response: We appreciate the reviewer’s comment regarding the impact of sample size and the risk of type II error. As recommended, we have revised the manuscript to address this limitation more explicitly and earlier in the Discussion section. We added clarification that while the study was adequately powered for exploratory analysis, it was not designed to detect subtle differences with high statistical precision, and that the absence of significant group-level differences may reflect insufficient power rather than a true lack of effect. Furthermore, to contextualize the non-significant findings, we have now included non-parametric effect size estimates using Cliff’s delta for key comparisons. For example, the effect sizes between the treatment and control groups were small (δ = 0.293 and 0.229 in the right caudal anterior cingulate cortex and left rostral middle frontal cortex, respectively), while large effects were observed between responders and non-responders (δ = 0.704 and 0.815 in the same regions). The limitation section has also been revised to reflect this point and to underscore the need for larger-scale studies to validate these preliminary findings. (page 3, line 135-138; page 8, line 240-247; page 10, line 350-354)
Method
Statistical analysis
In addition to p-values, non-parametric effect sizes were estimated using Cliff’s delta to quantify the magnitude of group differences in cortical thickness changes. Cliff’s delta was interpreted based on conventional thresholds: |δ| < 0.147 as negligible, < 0.33 as small, < 0.474 as medium, and ≥ 0.474 as large.
Discussion
This study demonstrated that anti-CGRP mAbs treatment itself did not lead to significant cortical thickness changes compared to oral preventive medications. While the study was adequately powered for exploratory comparisons and initial hypothesis generation, it was not designed to detect subtle differences with high statistical precision. This limitation should be considered when interpreting non-significant findings in between-group comparisons. Although the sample size was sufficient for exploratory analysis, we acknowledge that it may have limited the ability to detect subtle but clinically relevant cortical changes. The absence of statistically significant group-level differences may, therefore, reflect insufficient power rather than a true lack of effect.
Limitations
Seventh, while we applied non-parametric statistical tests and false discovery rate corrections, the small sample size and group stratification inevitably raise the possibility of type II errors. Effect size estimates using Cliff’s delta revealed small effects between the anti-CGRP treatment and oral treatment groups, suggesting potential biological relevance that may not have reached statistical significance due to limited statistical power.
- The control group consisted of patients taking a variety of oral preventive drugs, including medications with different mechanisms (e.g., topiramate, beta-blockers, and antidepressants). This heterogeneity may confound comparisons between groups, especially if some treatments have structural brain effects. Although the limitations section addresses this issue, the implications are not fully understood. I recommend expanding this discussion and, if possible, performing sensitivity analyses (e.g., excluding beta-blockers) or subgroup analyses based on drug class to assess the robustness of the findings.
Author response: We appreciate the reviewer’s comment. As noted, the control group included patients taking various oral preventives with distinct pharmacological mechanisms, which may influence brain structure differently. While this issue was acknowledged in the limitations section, we agree that the implications warrant further discussion. We have accordingly expanded the relevant paragraph in the Discussion to elaborate on the potential confounding effects of treatment heterogeneity and the challenges it presents for interpretation.
Regarding subgroup or sensitivity analyses, we agree that such analyses would be valuable in principle. However, the number of patients receiving each specific medication (e.g., beta-blockers, antidepressants) was too small to allow meaningful subgroup comparisons or statistically robust sensitivity analyses. We have added a sentence in the limitations section to clarify this point.
(page 8, line 248-256; page 10, line 356-361)
Discussion
An additional challenge in interpreting the between-group results is the heterogeneity of medications used in the oral treatment group. The oral treatment group included a variety of preventive agents, such as topiramate, beta-blockers, and antidepressants, each of which may exert distinct effects on brain structure. Although our results did not show significant cortical differences between the anti-CGRP and oral treatment groups, we cannot rule out the possibility that these pharmacological differences may have contributed to variability in structural outcomes. Future studies with more homogeneous control arms or medication-specific subgroup analyses will be necessary to better delineate treatment-related brain effects.
Limitations
Lastly, the small number of patients receiving each type of oral preventive medication limited our ability to perform subgroup or sensitivity analyses based on drug class. As a result, we could not assess the potential structural brain effects of individual drugs separately. This heterogeneity in the oral treatment group may represent a source of confounding and should be considered when interpreting the findings.
- The finding that patients who respond to anti-CGRP mAbs show cortical thinning in specific regions is intriguing, but interpretation remains speculative. It is unclear whether these changes reflect functional recovery, normalization of previously altered structures, or possible atrophy. The current discussion briefly references previous studies, but I believe this section would benefit from a more in-depth analysis of the possible mechanisms underlying the observed changes in cortical structure, particularly in pain-related regions such as the anterior cingulate and prefrontal cortex.
Author response: We agree that the mechanisms underlying the observed cortical thinning in responders remain unclear and merit more detailed discussion. We have therefore expanded the Discussion section to explore several possible interpretations, including functional normalization, cortical plasticity associated with symptom improvement, and the distinction between adaptive reorganization versus potential atrophy. We also included more in-depth reference to previous studies investigating structural and functional changes in pain-processing regions such as the anterior cingulate and prefrontal cortex in relation to migraine pathophysiology and treatment. (page 8, line 260-280)
Discussion
In contrast, cortical thickness reductions were observed only in responders to anti-CGRP mAbs treatment. GLM analysis further indicated that treatment response, rather than anti-CGRP mAbs administration itself, was significantly associated with cortical thickness reduction in specific brain regions. The cortical thinning observed in responders, especially in pain-processing regions such as the anterior cingulate and rostral middle frontal cortex, raises important questions regarding its biological significance. One plausible interpretation is that this reflects functional normalization of previously altered cortical areas. Several studies have reported that migraineurs exhibit cortical thickening in pain-related areas, likely due to ongoing nociceptive input and maladaptive neuroplastic changes [7,8]. Effective treatment may reduce this input and induce cortical reorganization toward a more normalized structure. Alternatively, these findings may represent adaptive plasticity, such as synaptic pruning or modulation of gray matter density associated with symptom relief, as observed in other chronic pain conditions [17]. It is important, however, to distinguish such dynamic reorganization from pathological atrophy, which typically accompanies neurodegeneration and is unlikely to occur over a short treatment duration in otherwise healthy individuals. Given the 3-month timeframe and lack of clinical deterioration in our cohort, atrophy is unlikely to explain the observed findings. Functional imaging studies have further shown that effective migraine treatment can modulate activity in the anterior cingulate and prefrontal cortex, regions involved in both pain affect and top-down modulation [13,18,19]. Our structural findings may reflect similar treatment-induced adaptations at the morphometric level. Nevertheless, further longitudinal and multimodal studies are needed to clarify whether the observed changes represent normalization, plasticity, or treatment-induced reorganization.
References
Kim JH, et al. Thickening of the somatosensory cortex in migraine without aura. Cephalalgia. 2014;34(14):1125-1133.
DaSilva AF, et al. Thickening in the somatosensory cortex of patients with migraine. Neurology. 2007;69(21):1990–1995.
Seminowicz DA, et al. Cognitive-behavioral therapy increases prefrontal cortex gray matter in patients with chronic pain. J Pain. 2013;14(12):1573–1584.
Schwedt TJ, et al. Longitudinal changes in functional connectivity and pain-induced brain activations in patients with migraine: a functional MRI study pre- and post- treatment with Erenumab. J Headache Pain. 2022;23(1):159.
Dai W, Liu R-H, Qiu E, et al. Cortical mechanisms in migraine. Molecular Pain. 2021;17. doi:10.1177/17448069211050246
Ofoghi Z, Rohr CS, Dewey D, et al. Functional connectivity of the anterior cingulate cortex with pain-related regions in children with post-traumatic headache. Cephalalgia Reports. 2021;4. doi:10.1177/25158163211009477
- The use of generalized linear models (GLMs) is appropriate, but given the small sample size, there is a high risk of overfitting. The authors should clarify how they checked assumptions (e.g., multicollinearity, residual analysis) and whether model selection strategies (e.g., AIC, BIC) were employed. Parsimonious modelling may be more appropriate in this context. Additionally, p values close to the significance threshold (e.g., p = 0.070) should be interpreted with caution and not overemphasized in the narrative.
Author response: We appreciate the reviewer’s comment. To minimize the risk of overfitting, we adopted a parsimonious modeling strategy based on clinical relevance and prior hypotheses, rather than stepwise selection procedures. Each GLM included only six predictors, and we confirmed that multicollinearity was not problematic (all VIFs < 2.0). Model fit and appropriateness were evaluated using multiple criteria. In the model for the left rostral middle frontal cortex, the deviance per degree of freedom was low (0.674/23), and information criteria values indicated good model fit (AIC = –12.735, AICC = –5.878, BIC = –1.526). In contrast, the model for the right caudal anterior cingulate cortex showed higher deviance (2.959/23) and less favorable AIC (31.646), AICC (38.503), and BIC (42.856) values, suggesting a more modest fit. Likelihood ratio tests for both models were not statistically significant (p = 0.232 and 0.368, respectively), and we have interpreted these results accordingly.
Regarding the p-value of 0.070 observed for the anti-CGRP treatment variable in the left rostral middle frontal cortex, we agree that such near-threshold results should be interpreted cautiously. We have revised the relevant sentence in the Discussion to report this value descriptively and to avoid overemphasis. (page 4, line, 145-151; page 7, line 213-226; page 8, line 258-260)
Methods
Subsequently, generalized linear models (GLMs) with a Gaussian distribution were used to identify clinical variables associated with changes in cortical thickness. The Shapiro-Wilk test confirmed that changes in cortical thickness followed a normal distribution. The dependent variable was the change in cortical thickness between baseline and follow-up MRI, while the candidate independent variables included anti-CGRP mAbs treatment, treatment response to anti-CGRP mAbs, age, body mass index, chronic migraine, and baseline MHDs. To avoid overfitting, the number of predictors was limited based on clinical relevance and pre-specified hypotheses. Stepwise selection procedures were not employed. Multicollinearity among predictors was assessed using variance inflation factors, with all values below 2.0. Residuals were visually inspected and showed no major violations of model assumptions. Model fit was evaluated using deviance per degree of freedom, Akaike information criterion (AIC), corrected AIC (AICC), Bayesian information criterion (BIC), and consistent AIC (CAIC).
Results
Generalized linear models identified significant associations between treatment response to anti-CGRP mAbs and cortical thickness changes in specific brain regions (Table 4). In the right caudal anterior cingulate cortex, treatment response to anti-CGRP mAbs exhibited significantly greater reduction of cortical thickness compared to non-responders (β=-0.429, 95% CI: -0.777 to -0.081, p = 0.016). The GLM for the right caudal anterior cingulate cortex showed a deviance per degree of freedom of 2.959, with AIC = 31.646, AICC = 38.503, BIC = 42.856, and CAIC = 50.856. Similarly, in the left rostral middle frontal cortex, responders showed significant reduction of cortical thickness (β=-0.224, 95% CI: -0.390 to -0.057, p = 0.008). The GLM for the left rostral middle frontal cortex demonstrated a deviance per degree of freedom of 0.674. Model fit indices were as follows: AIC = –12.735, AICC = –5.878, BIC = –1.526, and CAIC = 6.474. Other clinical variables, including age, BMI, chronic migraine status, baseline MHDs, and anti-CGRP treatment itself, were not significantly associated with cortical thickness changes in either region.
Discussion
GLM analysis further indicated that treatment response, rather than anti-CGRP mAbs administration itself, was significantly associated with cortical thickness reduction in specific brain regions.
- GLM analysis indicated that treatment response was significantly associated with cortical thickness reduction in specific brain regions, whereas anti-CGRP mAbs administration itself did not show a statistically significant association.
- The use of ATROSCAN for cortical thickness analysis is a novel element of the study, but its application and validation in this specific context are not sufficiently explained. FreeSurfer remains the most widely used tool in this field, so it is essential to briefly justify the choice of ATROSCAN, the reference validation studies and comment on its comparability in terms of reliability and sensitivity, particularly for detecting subtle changes in cortical morphology.
Author response: We appreciate the reviewer’s comment. ATROSCAN is a surface-based morphometry tool that has been validated against FreeSurfer, the most widely used software for cortical thickness analysis. In a prior study, ATROSCAN-generated cortical thickness values demonstrated a high level of agreement with FreeSurfer, with a reported Pearson correlation coefficient of 0.9623 (Lee at al., 2023). In addition, ATROSCAN incorporates data augmentation and advanced preprocessing steps that enhance segmentation robustness and reproducibility, even across varying MRI acquisition parameters. These features make it well suited for clinical datasets, and in our study, it enabled reliable regional cortical thickness measurement while minimizing preprocessing failures. We have revised the Methods section to clarify the rationale for using ATROSCAN and to provide supporting validation data. (page 3, line 121-126)
Reference
Lee, Suhyeon, et al. "Improving 3D imaging with pre-trained perpendicular 2D diffusion models." Proceedings of the IEEE/CVF International Conference on Computer Vision. 2023.
Methods
The ATROSCAN-generated cortical thickness measurements were validated against FreeSurfer, a widely used neuroimaging analysis tool, with a reported Pearson correlation coefficient of 0.9623, indicating high agreement between the two methods [17]. The robustness of ATROSCAN’s segmentation was further enhanced by applying data augmentation and preprocessing techniques to ensure reliable and reproducible cortical thickness measurements across different MRI acquisition settings.
Minor comments
- Abstract: Please clarify in the abstract that only significant cortical differences were observed among responders within the anti-CGRP group, and that no significant differences were found in the overall comparison with the control group.
Author response: Thank you for this helpful comment. We have revised the abstract according to the reviewer’s suggestions. (page 1, line 21-23)
Results of Abstract
Cortical thickness changes did not differ significantly between anti-CGRP and oral treatment groups. However, responders to anti-CGRP mAbs showed significant de-creases in cortical thickness compared to non-responders, particularly in the right caudal anterior cingulate (p = 0.026) and left rostral middle frontal cortex (p = 0.007).
- Cortical thickness changes did not differ significantly between the anti-CGRP and oral treatment groups. However, among patients receiving anti-CGRP mAbs, responders showed significant decreases in cortical thickness compared to non-responders, particularly in the right caudal anterior cingulate (p = 0.026) and left rostral middle frontal cortex (p = 0.007).
- Terminology: Ensure consistent use of terms such as “treatment group” versus “anti-CGRP group” throughout the manuscript.
Author response: Thank you for the helpful suggestion. We have revised the terminology to consistently use “anti-CGRP group” instead of “treatment group” and “oral treatment group” instead of “control group” when referring to the two study groups.
- Figures: Figure 2 would benefit from clearer anatomical labels and a legend indicating the direction and magnitude of cortical changes.
Author response: Thank you for the helpful suggestion. We have revised Figure 2 to include anatomical labels and a descriptive legend. Panel (A) now illustrates the cortical thickness difference between responders and non-responders in the right caudal anterior cingulate cortex and panel (B) shows the corresponding difference in the left rostral middle frontal cortex. The legend also indicates the direction and magnitude of cortical changes. We hope this improves clarity and interpretability for readers.
Figure 2. Cortical thickness changes in anti-CGRP responders versus non-responders. (A) The right caudal anterior cingulate cortex showing significant cortical thinning in responders compared to non-responders (B) The left rostral middle frontal cortex showing significant reduction in cortical thickness in responders. Brain surfaces are displayed with anatomical labels and group differences in cortical thickness are visualized.
- Language: There are several minor grammatical issues (e.g., “an increase” should be “an increase”) that should be addressed with a final revision of the language.
Author response: Thank you for the comment. We have carefully revised the manuscript for grammar and clarity throughout.

Reviewer 2 Report
Comments and Suggestions for Authors
Soohyun Cho's manuscript is devoted to the study of cortical thickness changes in migraine patients treated with anti-CGRP monoclonal antibodies. The manuscript is logical, the methods are described in detail, the figures and tables are informative and of good quality, and the data obtained are credible. At the same time, the manuscript has a moderate degree of novelty because anti-CGRP mAbs (3-month treatment) have previously been shown to cause reduction in anterior cingulate cortical thickness (Szabó et al.). A special feature of this paper is the analysis of 46 cortical regions in patients with migraine, although this is not reflected anywhere except in one phrase in the abstract and one in the materials and methods. It would be very useful to present data on the results of the analysis of all 46 cortical regions studied (at least in the supplementary materials). These data could be useful to other researchers. Also it would be useful to provide some MRI images to demonstrate the data obtained.
- Note 2022 in the heading of each page.
- This study was conducted as part of a clinical trial. If this trial is registered and there is information about it online, it would be good to cite it.
- It would be interesting to see in Discussion what the author thinks about be the reason for the individual susceptibility to anti-CGRP mAbs. Whether this is due to variability in the amino acid sequence of CGRP (e.g. due to SNPs in the CGRP gene) or to the fact of producing the neutralizing antibodies to anti-CGRP mAbs in non-responsive patients.
Author Response
Reviewer 2
Soohyun Cho's manuscript is devoted to the study of cortical thickness changes in migraine patients treated with anti-CGRP monoclonal antibodies. The manuscript is logical, the methods are described in detail, the figures and tables are informative and of good quality, and the data obtained are credible. At the same time, the manuscript has a moderate degree of novelty because anti-CGRP mAbs (3-month treatment) have previously been shown to cause reduction in anterior cingulate cortical thickness (Szabó et al.).
- A special feature of this paper is the analysis of 46 cortical regions in patients with migraine, although this is not reflected anywhere except in one phrase in the abstract and one in the materials and methods. It would be very useful to present data on the results of the analysis of all 46 cortical regions studied (at least in the supplementary materials). These data could be useful to other researchers.
Author response: Thank you for this valuable suggestion. We have added the full results of the cortical thickness analysis across all 46 brain regions in the supplementary materials. Specifically, we provide two separate tables: one comparing the anti-CGRP group with the oral treatment group, and another comparing responders and non-responders within the anti-CGRP group (Supplementary Tables 1 and 2).
- Also it would be useful to provide some MRI images to demonstrate the data obtained.
Author response: Thank you for the suggestion. We have added Supplementary Figure 1, which includes an overview of the cortical thickness analysis pipeline (ATROSCAN workflow) and a representative example of patient-specific cortical thickness data. This figure is intended to visually demonstrate the processing steps and types of data obtained in this study.
Supplementary Figure 1. (A) Workflow of cortical thickness analysis using ATROSCAN, including preprocessing, model prediction (gray matter segmentation and brain parcellation), and cortical thickness estimation. (B) Example of cortical thickness measurements visualized on a 3D rendered brain surface from a representative patient.
Note 2022 in the heading of each page.
Author response: Thank you for the comment. We have revised “2022” to “2025” on each page.
This study was conducted as part of a clinical trial. If this trial is registered and there is information about it online, it would be good to cite it.
Author response: Thank you for your comment. This study was approved by the Institutional Review Board; however, it was not registered as a clinical trial online.
It would be interesting to see in Discussion what the author thinks about be the reason for the individual susceptibility to anti-CGRP mAbs. Whether this is due to variability in the amino acid sequence of CGRP (e.g. due to SNPs in the CGRP gene) or to the fact of producing the neutralizing antibodies to anti-CGRP mAbs in non-responsive patients.
Author response: Thank you for this insightful comment. We have added a paragraph to the Discussion addressing potential biological mechanisms underlying individual variability in response to anti-CGRP mAbs. These include genetic differences such as polymorphisms in the CGRP gene that may affect ligand–receptor binding affinity, as well as the possible development of neutralizing antibodies that reduce the therapeutic efficacy of monoclonal antibodies. Although these factors remain speculative, they suggest the need for further research into predictors of treatment response and personalized approaches to migraine therapy. (page 9, line 312-318)
Discussion
The variability in individual response to anti-CGRP mAbs may be attributed to several biological factors. Genetic differences such as polymorphisms in the CGRP gene may influence ligand–receptor binding affinity. Additionally, the development of neutralizing antibodies against monoclonal antibodies could attenuate therapeutic efficacy in some patients. Although these mechanisms remain speculative, they underscore the need for further research to identify predictors of treatment response and to guide personalized migraine therapy.

Round 2
Reviewer 1 Report
Comments and Suggestions for Authors
Although the authors have adequately addressed all reviewer comments and the manuscript is suitable for publication, there are still some minor issues related to methodological reporting and consistency of language that need to be addressed before final acceptance.
- Although the Discussion addresses the power limitations well, the Abstract could benefit from a brief qualifier emphasizing the exploratory nature of the study and the small sample size.
- The Conclusion could briefly reiterate that the findings are preliminary owing to the small sample size and heterogeneity of oral treatments.
- Although the cover letter notes that the study was not registered as a clinical trial, it is better to explicitly state in the Methods section that “This study was not registered in a public trial registry.”
- Although most of the usages were corrected (e.g., “anti-CGRP group” vs. “treatment group”), do a final sweep for consistency in the Results and Tables legends (some inconsistent usages persist, such as “oral treatment treatment group” on page 5).
- Minor problems remain (e.g., repetitions in some sentences, awkward expressions such as “In contrastInstead” and inconsistent use of past/present). A final review by a native speaker or editor would be helpful.
- Check that Supplementary Tables 1 and 2 and Figure 1 are correctly cited in the main text, and that their file format, numbering, and legends meet journal standards.
- Some references (e.g., Lee et al., 2023; Amaral et al., 2018) may need formatting adjustments to align with the journal's citation style.
Author Response
Dear editor and reviewers,
We are grateful for the constructive comments. We would like to thank the editor and reviewers for their time and effort to review our manuscript. We made our best efforts to respond to all concerns raised by the reviewers.
Reviewer 1:
Although the authors have adequately addressed all reviewer comments and the manuscript is suitable for publication, there are still some minor issues related to methodological reporting and consistency of language that need to be addressed before final acceptance.
- Although the Discussion addresses the power limitations well, the Abstract could benefit from a brief qualifier emphasizing the exploratory nature of the study and the small sample size.
Author response: Thank you for the comment. We have revised the Abstract to briefly note the exploratory nature of the study and the small sample size, as suggested.
Conclusion of Abstract
Anti-CGRP mAbs treatment itself did not lead to region-specific cortical thickness changes compared to oral preventives. However, treatment response to anti-CGRP mAbs was associated with region-specific cortical thickness changes. These findings suggest that anti-CGRP mAbs may modulate migraine-related brain abnormalities in responders, providing insights into their central mechanisms of action beyond peripheral effects.
- This exploratory study, based on a small sample size, suggests that cortical thickness changes may be associated with treatment response to anti-CGRP mAbs rather than with CGRP mAb treatment itself. Further studies with larger cohorts are needed to confirm these findings.
- The Conclusion could briefly reiterate that the findings are preliminary owing to the small sample size and heterogeneity of oral treatments.
Author response: Thank you for the suggestion. We have revised the Conclusion to clarify that the findings are preliminary, reflecting the small sample size and heterogeneity of the oral treatment group.
Conclusion
This exploratory study suggests that cortical thickness changes may be more closely associated with treatment response to anti-CGRP mAbs than with the administration of anti-CGRP mAbs themselves. These findings suggest that anti-CGRP mAbs may modulate migraine-related brain abnormalities in responders, providing insights into their central mechanisms of action beyond peripheral effects. However, given the small sample size and the heterogeneity of oral preventive treatments used in the control group, these findings should be considered preliminary and require validation in larger, more homogeneous cohorts.
- Although the cover letter notes that the study was not registered as a clinical trial, it is better to explicitly state in the Methods section that “This study was not registered in a public trial registry.”
Author response: Thank you for the comment. As suggested, we have added a sentence to the Methods section to clarify that this study was not registered in a public trial registry.
Methods
This study was approved by the Institutional Review Board of Uijeongbu Eulji Medical Center (IRB No. EMC 2021-10-009), and all participants provided written informed consent. This study was not registered in a public trial registry.
- Although most of the usages were corrected (e.g., “anti-CGRP group” vs. “treatment group”), do a final sweep for consistency in the Results and Tables legends (some inconsistent usages persist, such as “oral treatment treatment group” on page 5).
Author response: Thank you for pointing this out. We have carefully reviewed the manuscript and corrected all remaining inconsistencies in terminology, including the repetition error on page 5.
- Minor problems remain (e.g., repetitions in some sentences, awkward expressions such as “In contrastInstead” and inconsistent use of past/present). A final review by a native speaker or editor would be helpful.
Author response: Thank you for the careful review. We thoroughly re-checked the manuscript for typographical errors, repetitive phrasing, awkward expressions, and tense inconsistencies.
- Check that Supplementary Tables 1 and 2 and Figure 1 are correctly cited in the main text, and that their file format, numbering, and legends meet journal standards.
Author response: Thank you for your comment. All supplementary materials have been compiled into a single PDF file according to the journal’s formatting guidelines. We also revised the legends for clarity and consistency and adjusted the layout and labels to improve readability and alignment with journal standards.
Methods
The processing pipeline and representative patient-level cortical maps are presented in Supplementary Figure S1.
Results
A full list of regional cortical thickness changes is provided in Supplementary Tables S1 and S2.
- Some references (e.g., Lee et al., 2023; Amaral et al., 2018) may need formatting adjustments to align with the journal's citation style.
Author response: Thank you for pointing this out. We have reviewed and revised the formatting of references to ensure consistency with the journal’s citation style.
